# Predicted structure of fully activated human bitter taste receptor TAS2R4 complexed with G protein and agonists

Moon Young Yang[1] ⓘ, Amirhossein Mafi[1] ⓘ, Soo-Kyung Kim[1] ⓘ, William A. Goddard III[1]* ⓘ and Brian Guthrie[2] ⓘ

[1]Materials and Process Simulation Center, California Institute of Technology, Pasadena, CA 91125, USA and [2]Cargill Global Food Research, Wayzata, MN 55391, USA

## Research Article

**Key words:**
G protein-coupled receptor; GEnSeMBLE method; GPCR; gustducin G protein; metadynamics; molecular dynamics

**Author for correspondence:**
*William A. Goddard III,
E-mail: wag@caltech.edu

## Abstract

Bitter taste is sensed by bitter taste receptors (TAS2Rs) that belong to the G protein-coupled receptor (GPCR) superfamily. In addition to bitter taste perception, TAS2Rs have been reported recently to be expressed in many extraoral tissues and are now known to be involved in health and disease. Despite important roles of TAS2Rs in biological functions and diseases, no crystal structure is available to help understand the signal transduction mechanism or to help develop selective ligands as new therapeutic targets. We report here the three-dimensional structure of the fully activated TAS2R4 human bitter taste receptor predicted using the GEnSeMBLE complete sampling method. This TAS2R4 structure is coupled to the gustducin G protein and to each of several agonists. We find that the G protein couples to TAS2R4 by forming strong salt bridges to each of the three intracellular loops, orienting the activated Gα5 helix of the Gα subunit to interact extensively with the cytoplasmic region of the activated receptor. We find that the TAS2Rs exhibit unique motifs distinct from typical Class A GPCRs, leading to a distinct activation mechanism and a less stable inactive state. This fully activated bitter taste receptor complex structure provides insight into the signal transduction mechanism and into ligand binding to TAS2Rs.

## Introduction

Bitter taste perception is generally assumed to protect an organism against potentially toxic substances that are sensed by bitter taste receptors (TAS2Rs). TAS2Rs belong to the G protein-coupled receptors (GPCRs) superfamily with 25 expressed in humans. The TAS2Rs interact with a large and diverse group of compounds. Surprisingly, TAS2Rs are expressed not only in the oral cavity but also in many extraoral tissues, including gastrointestine (Wu *et al.*, 2002), nasal epithelium (Finger *et al.*, 2003), airway (Deshpande *et al.*, 2010), skin (Shaw *et al.*, 2018), heart (Foster *et al.*, 2014) and brain (Singh *et al.*, 2011*a*). It has been reported that TAS2Rs expressed in extraoral tissues are deeply involved in health and disease by contributing to physiological functions, such as sensing potentially harmful molecules and regulating the metabolic/immune system (Lu *et al.*, 2017). To date, a number of studies have reported about TAS2Rs-associated disorders and diseases including food intake (Andreozzi *et al.*, 2015), insulin homeostasis (Dotson *et al.*, 2008), cardiovascular disease (Shiffman *et al.*, 2008) and cancer (Martin *et al.*, 2019).

Despite crucial roles in physiological functions and related diseases, investigations on TAS2Rs have been hampered by the lack of structural information. Neither X-ray crystal nor cryo-electron microscopy (EM) structures are available for TAS2Rs. Thus the only structural information is from several predicted structures based mostly on homology modelling using Class A GPCRs (Singh *et al.*, 2011*b*; Di Pizio & Niv, 2015; Pydi *et al.*, 2015; Pándy-Szekeres *et al.*, 2018). Such homology models for GPCRs are problematic because the seven transmembrane domains (TMD) form nearly parallel α-helices so that their axial orientation is highly sequence dependent. Moreover, although TAS2Rs are generally classified as Class A GPCRs, the sequence similarity of TAS2Rs with other class A GPCRs is much less than 30% (Di Pizio *et al.*, 2016) making homology modelling even less reliable. To ameliorate this problem we developed a systematic approach to take a GPCR with a known structure as a template to fix the initial tilts for the seven TMDs and then sampling first all coupled axial rotations (BiHelix step, 35 million combinations) (Abrol *et al.*, 2012) and then various combinations of tilts and rotations (SuperBiHelix, 13 trillion combinations) (Bray *et al.*, 2014) combined with side chain optimization (SCREAM) to finally select the lowest 25 conformation likely to play a role in ligand binding. This methodology, referred to as the GPCR ensemble of structures in membrane bilayer environment (GEnSeMBLE), has been successful for predicting other GPCRs, including Class A [CCR5 (Berro *et al.*, 2013; Abrol *et al.*, 2014), AA3 (Goddard *et al.*, 2010), DP prostaglandin (Shankar *et al.*, 2018) and CB1 (Ahn *et al.*, 2013; Scott *et al.*, 2013)], Class B [GLP1 (Kirkpatrick *et al.*,

2012)] and Class C [TAS1R2/1R3 sweet heterodimer (Kim *et al.*, 2017)], but this is the first application of this methodology to TAS2Rs [an earlier application of some of these tools was used to predict the structure of TAS2R38 (Tan *et al.*, 2012)].

Having the structure for the GPCR is just the first step. To describe the signal transduction we must consider how binding of agonist and G protein (GP) to the GPCR leads to activation. We find that binding an agonist to the GPCR-GP complex results in a conformational change of the GPCR that is transmitted to the bound GP, leading to opening the tight GP-GDP complex to form the activated state with an open Gα subunit. In our recent report for κ opioid receptor (κOR), we found that the $G_i$ protein ($G_iP$) makes strong anchors to all three intracellular loops (ICLs) of the κOR, stabilizing the active conformation of GPCR (Mafi *et al.*, 2020*a*, 2020*b*). This indicates that both ligand and GP interacting with the GPCR lead to structural changes that stabilize the final activated state. Therefore, to understand the factors responsible for attaining the fully activated GPCR-agonist-GP complex structure, we need to understand the role of these three components. This should help develop a structural basis for developing new more selective and active agonists or antagonists.

In this study, we describe the prediction of the three-dimensional (3D) structure of the fully activated TAS2R4 human bitter taste receptor with several agonists using the GEnSeMBLE (Abrol *et al.*, 2012; Bray *et al.*, 2014) and DarwinDock (Griffith, 2017) complete sampling methods. We find that the agonist binding modifies the hydrogen-bond (HB) network in the binding pocket, which triggers a shift in the conformation of $Y^{6.48}$, which probably acts as a transmission switch similar to that observed in typical Class A GPCRs. Moreover, we find that the GP, gustducin heterotrimeric protein (responsible for the bitter and basic taste; hereafter referred to as $G_{gust}$), binds to the TAS2R4 by forming salt bridge (SB) anchors to each of the three ICLs of TAS2R4, orienting the Gα5 helix to make extensive interactions with residues in the cytoplasmic region of TAS2R4. We also find several unique motifs conserved only across TAS2Rs that result in an activation mechanism distinct from that of typical Class A GPCRs. The structure of this fully activated TAS2R4-agonist-$G_{gust}$ complex provides information that should be useful in understanding universal and diverse features across other TAS2Rs and for other subfamilies of GPCRs.

## Results and discussion

### Structural prediction for TAS2R4

Since no crystal structure is available for TAS2Rs, we predicted the 7-helix TMD of TAS2R4 using the GEnSeMBLE complete sampling technique. This method has successfully predicted 3D structures of many GPCRs (Goddard *et al.*, 2010; Abrol *et al.*, 2012; Kirkpatrick *et al.*, 2012; Tan *et al.*, 2012; Ahn *et al.*, 2013; Berro *et al.*, 2013; Scott *et al.*, 2013; Abrol *et al.*, 2014; Bray *et al.*, 2014; Kim *et al.*, 2017; Shankar *et al.*, 2018). The first step in GEnSeMBLE is to select a template from known structures to specify the initial tilts of the seven TMDs. Despite its low sequence similarity, most analyses support the classification of TAS2Rs with Class A GPCRs (Di Pizio and Niv, 2015; Di Pizio *et al.*, 2016). Therefore, based on the sequence alignment for Class A GPCRs for which active structures are available, we selected four templates with highest homology to TAS2R4 for the GEnSeMBLE predictions (Table S1)); serotonin 2C (5-HT2C), angiotensin 2 (AT2), cannabinoid 1 (CB1) and rhodopsin. For each template we mapped the residues to TAS2R4 and

examined coupled 30° rotations about the tilt axis of all seven TMDs. This BiHelix step examines $12^{**}7 = 35$ million combinations from which we select 2,000 that we build into 7-helix bundles while reoptimizing side chain conformations. From these 2,000 conformations we examined the 10 lowest energy shown in Table S2. We then selected two structures based on energy and diversity for the SuperBiHelix step that considers changing the tilt polar angle ($\theta$) by ±10°, the tilt azimuthal angle ($\phi$) by ±15°, ±30° and each axial angle ($\chi$) by ±15°, ±30° for a total of $(3 \times 5 \times 5)^{**}7 = 13$ trillion conformations. For each of these two cases, we select the 2,000 from SuperBiHelix predicted to have the lowest energy that we build into 7-helix bundles while reoptimizing side chain conformations by SCREAM. From this $2 \times 2,000$ we selected the best 25 by energy and diversity, as shown in Table S3, for docking each agonist. From these 25 we finally selected the energetically most favourable structure obtained from the 5-HT$_{2C}$ template for further study (Fig. S1).

### Ligand binding

#### Ligand binding site

Over 30 compounds are known to activate TAS2R4 (Wiener *et al.*, 2012). Of these, 10 ligands show binding constants of 100 μM effective concentration or stronger. Many of these ligands are not-specific to TAS2R4 (Table S4). Rubusoside (Rubu), a steviol glycoside isolated from the Chinese sweet tea plant *Rubus suavissimus*, is known to bind to TAS2R4 with 50 μM effective concentration (and to TAS2R14 with 400 μM effective concentration). Although steviol glycosides are generally known as nonnutritive sweeteners, they elicit a lingering bitter aftertaste that is mediated by TAS2R4 and TAS2R14 (Hellfritsch *et al.*, 2012). Thus, we focus on Rubu as an agonist of TAS2R4, which we compare to quinine, a well-known bitter compound that binds to several bitter receptors, including TAS2R4.

We used DarwinDock to predict the binding poses of Rubu and quinine to the most favourable 7-TMD conformation of TAS2R4 predicted by GEnSeMBLE (Fig. S2). DarwinDock considers ~50,000 poses of the ligand in the protein binding site and selects the top 100 by energy. In this process the six hydrophobic residues (I, L, V, F, Y and W) are replaced by Ala to allow space for sampling, but after selecting the top 100 poses by energy, we use SCREAM to add back these hydrophobic residues independently to each pose to provide each ligand pose a unique environment (Li *et al.*, 2015). Then we select the lowest energy poses for further analysis.

We find that Rubu forms HBs to one residue in TM3 ($D92^{3.36}$) and two in TM7 ($K262^{7.35}$ and $S263^{7.36}$) (the superscript is Ballesteros–Weinstein GPCR numbering (Ballesteros and Weinstein, 1995)). Charged (or polar) residues in TM3 of the binding pocket often play a crucial role in the ligand recognition and binding for Class A GPCRs, such as $D^{3.32}$ for β2 adrenergic receptor (β2AR) and all opioid receptors. Similarly, we find that $D92^{3.36}$ plays an important role in Rubu binding to TAS2R4, as discussed in detail below. This $D92^{3.36}$ in TAS2R4 is unique among TAS2Rs, where most TAS2Rs have asparagine at this position (Fig. S3). Thus $D92^{3.36}$ may play an important role in selectivity for TAS2R4.

Quinine is known to have a protonated nitrogen at physiological conditions (Thompson *et al.*, 2007), and our docking studies find that it makes a SB with $D92^{3.36}$. However, quinine is known to bind to nine other TAS2Rs that do not have a negatively charged residue at position 3.36, yet they lead to similar or even higher binding affinity (Wiener *et al.*, 2012). Thus we expect that formation of this SB is not likely to be essential for binding of quinine and may not induce a higher binding affinity. In addition to this interaction with

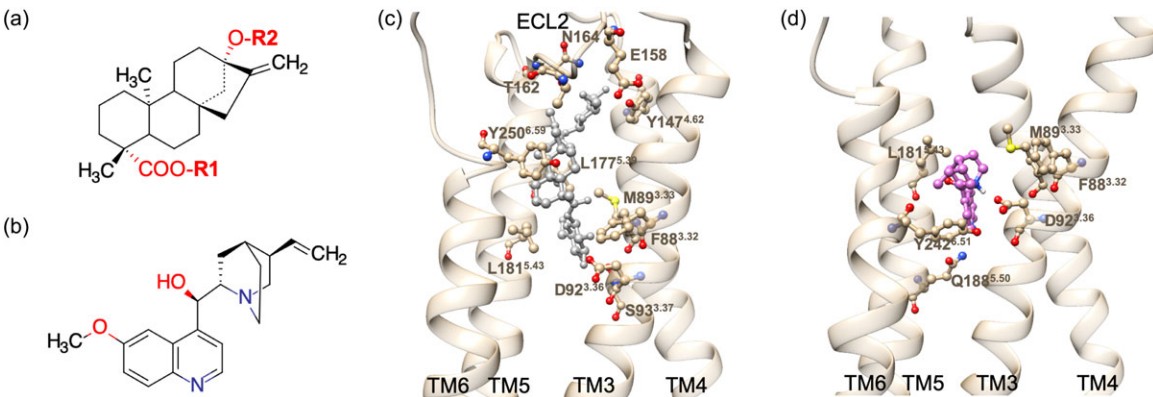

**Fig. 1.** Chemical structures of (*a*) the steviol glycosides and (*b*) quinine, where sugar (glucose or rhamnose) is attached at R1 and R2 in (*a*). Binding sites for (*c*) Rubu and (*d*) quinine to TAS2R4. Rubu has HBs to residues in TM3 (F88$^{3.32}$, M89$^{3.33}$, D92$^{3.36}$, S93$^{3.37}$), TM4 (Y147$^{4.62}$), TM5 (L177$^{5.39}$, L181$^{5.43}$), TM6 (Y250$^{6.59}$) and ELC2 (E158, T162, N164). Quinine has a SB to D92$^{3.36}$ and HBs to residues in TM3 (F88$^{3.32}$, M89$^{3.33}$), TM5 (L181$^{5.43}$, Q188$^{5.50}$) and TM6 (Y242$^{6.51}$).

D92$^{3.36}$, quinine forms HBs to residues in TM5 (S184$^{5.46}$) and TM6 (Y242$^{6.51}$). These interactions of Rubu and quinine with TAS2R4 obtained from docking evolved during MD simulations after inserting in lipid bilayer and water box, as discussed below.

Steviol glycosides have a steviol as a core structure, containing a carboxyl hydrogen (R1 side) and a hydroxyl hydrogen (R2 side) at each end, with R1 and R2 replaced by various sugars (glucose or rhamnose) as shown in Fig. 1*a*. Rubu is the smallest among steviol glycosides, with just a glucose ring at each end of the core structure (Table S5). During the MD simulations with full lipid bilayer and solvent for 480 ns, Rubu made interactions with residues in TM3, TM4, TM5, TM6 and extracellular loop 2 (ELC2) (Fig. 1*c*, S4A and Table S6). Due to the hydrophobic core and the hydrophilic sugar rings of Rubu, residues in the middle of the binding pocket and close to the steviol core contribute to the binding mainly through hydrophobic interactions, whereas those close to the sugar rings are involved in HBs and polar interactions. The MD simulations were carried out with the GP bound (TAS2R4-agonist-G$_{gust}$); interactions between GPCR and G$_{gust}$ are discussed in section 'TAS2R4 in complex with the full heterotrimeric gustducin G protein.'

In comparison, quinine is relatively small, interacts with fewer residues in TM3, TM5, and TM6 (Fig. 1*b,d*, S5A and Table S6). Van der Waals or hydrophobic interactions dominate, except for the SB interaction of the quinine tertiary nitrogen with D92$^{3.36}$.

In addition to Rubu, other bulkier steviol glycosides can also activate TAS2R4 (Hellfritsch *et al.*, 2012), although experimental result shows that those are not as bitter as Rubu (Table S5). For example, Rebaudioside M (RebM) has six sugar rings (three on each side of the core) and exhibits moderate bitterness. To investigate how such bulkier ligands bind to TAS2R4 and their activity (bitterness), we predicted an initial binding pose of RebM by matching to that of Rubu. This is because its large size complicates getting accurate results directly from docking. Then we performed MD simulation to equilibrate RebM in the binding site. We find that RebM makes more interactions with residues in TM2, TM7 and ECL1, compared to Rubu (Figs S6A, S6B and Table S6). It has been reported that TM1, TM2 and TM7 are more involved in binding of antagonists (Di Pizio *et al.*, 2016), which may be a reason for RebM being much less bitter than Rubu. Although the six sugar rings of RebM lead to a number of degrees of freedom with many possible conformations of the sugar rings that might lead to a different binding mode, the TAS2R4-RebM-G$_{gust}$ complex seems converged after 480 ns of MD simulations (Fig. S6C).

A derivative of RebM, hydRebM, exhibits much less bitterness, despite a very similar structure (Table S5). The only difference with RebM is that an alkene group in the core structure is substituted with hydroxyl and methyl groups. In comparison, isoRebM, in which the double bond of the alkene is shifted to the next carbon in the ring, shows bitterness similar to that of RebM (Table S5). These dramatic differences in activation for similar structures indicate that the hydroxyl group in hydRebM must play a role in suppressing activation. Indeed as described below our MD simulations show that this hydroxyl group makes an internal HB with one of R2 sugars, which may hinder the proper positioning of the ligand for binding and activation (Figs S7 and S8B). This hydroxyl group also makes an HB with Y250$^{6.59}$, which may lead to a change in the hydRebM binding pose or it may act to restrict the structural change of TM6 for activation, but this interaction was not retained during the dynamics (Fig. S8C). For the Rubu and RebM systems, we found that Y250$^{6.59}$ contributes to the binding, but mainly through hydrophobic interactions (Table S6). To fully understand the difference in bitterness across steviol glycosides, it may be necessary to investigate the transition in which the GP induces the inactive state to form the fully active state.

### The transmission switch of Y239$^{6.48}$

Among the residues in the binding pocket, D92$^{3.36}$ likely plays a particularly crucial role in ligand binding. It forms two stable HBs with Rubu and a SB with quinine, contributing to almost ¼ and ½ of the total binding, respectively (Fig. 1, S4C and S5C and Table S6). Not only does D92$^{3.36}$ provide a large energy stabilization *via* polar interactions, but it contributes to transferring the signal for activation by changing the HB network (Fig. 2). Thus in the apo protein structure from GEnSeMBLE, we find that D92$^{3.36}$ makes an HB interaction with Y239$^{6.48}$. But the presence of agonist causes the side chain of Y239$^{6.48}$ to break this interaction with D92$^{3.36}$ so that it rotates anti-clockwise. Indeed, it is well-known that the highly conserved W$^{6.48}$ in Class A GPCRs plays a crucial role in activation, where it is known as a 'transmission switch' (Zhou *et al.*, 2019). Although Y239$^{6.48}$ is not as highly conserved in TAS2Rs as is W$^{6.48}$ in typical Class A GPCRs, this significant movement of Y239$^{6.48}$ upon ligand binding suggests that this conformational change may contribute to the rotation and structural change in TM6, which is one of most significant changes between inactive and active states observed in Class A GPCRs.

To validate this hypothesis, we prepared two structural models without an agonist. The first model (model 1) is the one constructed

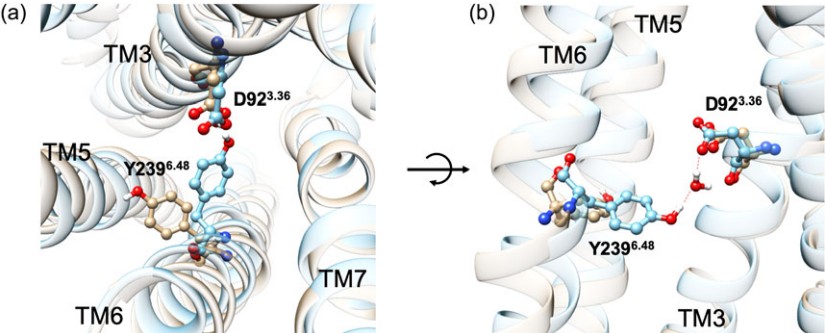

**Fig. 2.** (*a*) Top and (*b*) side views showing $D92^{3.36}$ – $Y239^{6.48}$ interactions in the absence (cyan) and presence (tan) of the agonist, but the agonist is omitted for clarity.

by GEnSeMBLE (Fig. S1), where it was equilibrated with neither agonist nor $G_{gust}$, although this model was constructed using an activated GPCR as a template. The second model (model 2) is the pre-activated structure, which contains no agonist but is coupled to the inactive $G_{gust}$ coupled to GDP. This is motivated by our discovery that GPs form anchors that induce the conformational change of the receptor (Mafi *et al.*, 2020*a*) discussed in the following section. This led to our G protein-first hypothesis, in which GP binds first to the receptor to initiate activation followed by agonist binding to complete activation. Both models find that $Y239^{6.48}$ maintains a stable direct or water-mediated HB with $D92^{3.36}$ for 100 ns of MD simulation (Fig. S9A). A further metadynamics simulations (metaMD) based free energy analysis shows an energy minimum corresponding to a water-mediated HB between $D92^{3.36}$ and $Y239^{6.48}$ (Fig. S9B). Thus, $Y239^{6.48}$ and $D92^{3.36}$ play crucial roles in the activation process, just as does $W^{6.48}$ for typical Class A GPCRs.

We attribute the decreased bitterness of RebM and hydRebM relative to Rubu to the interaction of these ligands with $Y239^{6.48}$. Compared to Rubu, both RebM and hydRebM have two additional sugar rings attached to the first sugar ring at the R1 side, and one of these sugars can reach $Y239^{6.48}$ to make an HB (Fig. S10). This additional interaction may hinder the conformational change of $Y239^{6.48}$ needed for activation, resulting in suppressing activation. A similar interaction may also account for the lower bitterness observed in other bulky steviol glycosides (Table S5).

### TAS2R4 complexed with the full heterotrimeric gustducin G protein

Although structures for many GPCRs with ligand binding sites have been predicted successfully using GEnSeMBLE and Darwin-Dock, it remains a challenge to predict a fully activated structure with decisive information for understanding the signal transduction mechanism and for developing new therapeutics. This is because the activation process requires binding of *both* agonist *and* the cognate GP. Both components are usually essential for activating GPCRs to induce subsequent signal transduction. In our recent MD and metaMD studies of κOR and μOR, we discovered that these GPCRs binds strongly to its cognate GP *via* SB and HB interactions at all three ICLs (Mafi *et al.*, 2020*a*). These interactions were not identified in the cryo-EM structures because of the low resolution and disorder. To obtain this fully activated structure for TAS2R4, we used the lessons from κOR and μOR to predict the complex structure. Here we used the gustducin heterotrimeric protein, which is responsible for the bitter taste.

Indeed, we found essentially the same interactions for the TAS2R4-$G_{gust}$ complex as for $G_iP$ with κOR and μOR. Fig. 3*a* shows

the equilibrated TAS2R4-Rubu-$G_{gust}$ structure immersed in the lipid bilayer. For the 480 ns MD simulation, the root mean square deviation (RMSD) of the TAS2R4 was within 3 Å while that of the whole protein complex was 4 Å, showing the stability of the complex structure (Fig. S4B).

- For anchor 1, we find that two positive residues in ICL1 form SBs with two negative residues in the Gβ subunit, K37ICL1 – D312Gβ and R40ICL1 – D333Gβ (Fig. 3*b*).
- For anchor 2, we find that R123 in ICL2 forms SBs with both E25Gα and E28Gα of the Gα subunit (Fig. 3*c*).
- For anchor 3, we find that two lysine residues in ICL3 interact with both the Gα5 helix and with the Ras-like domain of the Gα subunit, K209(ICL3) – D341(D337)Gα and K212ICL3 – E318 (D316)Gα (Fig. 3*d*).

These residues involved in anchor formation were well positioned when we superimposed the mouse μOR-$G_iP$ structure with TAS2R4-$G_{gust}$ during the model construction step (Fig. S11). A free energy analysis using metaMD shows that these SBs should form spontaneously with activation of $G_{gust}$ as it approaches TAS2R4, stabilizing the structural complex (Fig. S12). The three anchors formed between TAS2R4 and $G_{gust}$ were quite stable over ~0.5 μs MD simulation, although occasional fluctuations and switching of SB pairs were observed (Fig. S4D–F).

We found similar interactions between TAS2R4 and $G_{gust}$ for the TAS2R4-quinine-$G_{gust}$ complex structure (Fig. S5), and also for the RebM and hydRebM systems (Figs S6 and S7). Therefore, we conclude that this formation of anchors between $G_{gust}$ and the ICLs of the GPCR is likely a common feature across all TAS2Rs and Class A GPCRs.

It is well known that activation of the GPCR-agonist-GP complex involves coupling of the Gα5 helix of the Gα subunit with residues deep in the GPCR (Onrust *et al.*, 1997; Oldham *et al.*, 2006; Oldham & Hamm, 2008). Indeed for κOR and μOR we found that Gα5 forms a network of HBs and SBs to the GPCR (Mafi *et al.*, 2020*a*). For these opioid receptors, a conserved arginine residue in ICL2 (R170 for κOR and R179 for μOR respectively) plays an important role in the interaction with Gα5, forming a SB with D350Gα5. For TAS2R4, we find that the highly conserved $K109^{3.53}$ over all TAS2Rs plays the same role, forming a SB with D350Gα5 (Fig. 3*e*).

For κOR and μOR the terminal carboxylate of Gα5 (F354) is also important in proper positioning of the Gα5 helix, making a SB with $R^{6.32}$ in TM6 in the fully activated structure. For most TAS2Rs, we expect an analogous interaction because most TAS2Rs have a conserved positively charged residue in TM6 (6.32 or 6.36 position)

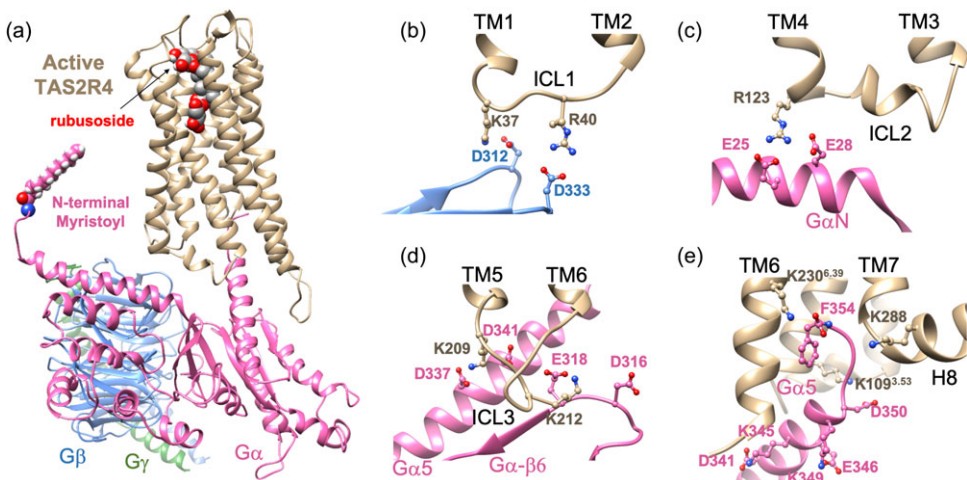

**Fig. 3.** (*a*) The structure of TAS2R4-Rubu-G$_{gust}$ complex. Important polar interactions at three anchors: (*b*) anchor 1: ICL1 with Gβ, (*c*) anchor 2: ICL2 and GαN and (*d*) anchor 3: ICL3 and Gα. (*e*) Polar interactions between the Gα5 helix of the Gα subunit with TAS2R4.

(Fig. S3). However TAS2R4 does *not* have a positively charged residue at either the 6.32 or 6.36 position. Therefore, we expect that F354Gα5 would make a SB with an alternative neighboring positive residue in TAS2R4. Our predicted structure leads to two candidates for this interaction, K230$^{6.39}$ and K288H8, both of which are well conserved (Figs 3*e* and S3). To test these possibilities we first constructed a K230$^{6.39}$ – F354Gα5 pair and carried out the MD simulation, finding that it changed to the K230$^{6.39}$ – K288H8 pair (Fig. S4G). For the TAS2R4-quinine-G$_{gust}$ complex structure, F354Gα5 formed a SB with both of these lysine residues, alternating during the MD (Fig. S5G). For the RebM system F354Gα5 formed a stable SB with K230$^{6.39}$, whereas hydRebM shows an alternative interaction of F354Gα5 with either K230$^{6.39}$ or K288H8 (Figs S6H and S7H). Although it is unclear whether this interaction should depend on the ligand, we consider it likely that the terminal carboxylate of Gα5 makes a SB with either or both of K230$^{6.39}$ and K288H8 in TAS2R4.

We expect that the final position of the Gα5 helix in the activated structure may be achieved in stages. There are reports that there is a stepwise sequential interactions of the Gα5 helix from its first contact toward a deeper insertion that induces opening the α-helical domain and GDP release (Du *et al.*, 2019). Indeed, we find that bringing the GDP bound G$_{gust}$ towards the GPCR leads to an initial SB contact of F354Gα5 with the highly conserved K109$^{3.53}$ (Fig. S13). As the GP is activated the Ras-like domain and the α-helical domain of the Gα subunit sandwiched around the GDP open up so that GDP is exposed to exchange with GTP. Thus in the activation mechanism, the Gα5 helix first interacts with the receptor through the F354Gα5 – K109$^{3.53}$ SB interaction, but then the helix gradually inserts more deeply upon activation. During this activation process, D350Gα5 makes a new SB with K109$^{3.53}$ while F354Gα5 forms a SB with K230$^{6.39}$ (or K288H8) of TAS2R4. This triggers the opening of the Gα and eventual GDP-GTP exchange signaling. Interestingly, even for the structural model with the inactive G$_{gust}$ (tightly coupled to the GDP), residues involved in the anchor formation were placed sufficiently close enough to make SBs (Fig. S13C–E). This suggests that the anchors form first, to help to position the Gα5 helix for the insertion. In our previous study of the κOR and μOR in which we discovered the formation of GPCR-G$_i$P anchors that induce conformational changes in the receptor, we suggested the 'G protein-first' paradigm, in which GP binds first to the receptor to initiate activation, followed by agonist binding to further open the cytoplasmic region of the receptor to complete the activation with GDP-GTP exchange (Mafi *et al.*, 2020*a*). We suggest that this applies to the TAS2Rs also.

### Conserved motifs in TAS2Rs

#### The 1-2-7 interactions

Although TAS2Rs are generally considered to belong to Class A, this classification of TAS2Rs is somewhat ambiguous due to the low sequence similarity. Moreover, it is known that TAS2Rs have a quite different pattern of conserved motifs compared to typical Class A GPCRs (Di Pizio *et al.*, 2016). For example typical Class A GPCRs have highly conserved N$^{1.50}$, D$^{2.50}$ and N$^{7.49}$ in the NPxxY motif that can coordinate a sodium ion (or protonated H$_2$O) and is known as the sodium binding pocket (Liu *et al.*, 2012). Based on high-resolution X-ray structures, the sodium ion is coordinated to the highly conserved D$^{2.50}$ and S$^{3.39}$ sites plus water molecules, stabilizing the inactive conformation. This sodium ion binding pocket is collapsed by the structural changes in the GPCR accompanying the activation process. For TAS2Rs, however, the aspartate that plays a critical role in coordination of the sodium ion for Class A is replaced with highly conserved arginine (Fig. S3). Thus, a sodium ion is unlikely to bind at this site as observed in typical Class A GPCRs. For TAS2Rs, mutagenesis studies show that the S$^{7.50}$A mutation induces hyperactivity (Pydi *et al.*, 2012), leading Pydi *et al.* to propose that S$^{7.50}$ stabilizes the inactive state by forming an HB with R$^{2.50}$. However our MD simulations find that in the active state, R55$^{2.50}$ forms an HB network with N24$^{1.50}$, H276$^{7.49}$ and S277$^{7.50}$ (Fig. 4*a*). We found that this HB network is stable and maintained for the full ~0.5 µs MD simulation (Fig. S14A–D). Moreover, our metaMD shows that this HB network is energetically favourable (Fig. S14E). All residues involved in this TM1–2–7 interaction are highly conserved, indicating that this HB network plays an important role in the activation process for TAS2Rs.

On the other hand, for the inactive structure constructed using GEnSeMBLE based on the inactive 5-HT$_{2C}$ template and then equilibrated (Table S7 and S8), we finds that R55$^{2.50}$ makes an HB with S95$^{3.39}$, instead of N24$^{1.50}$, H276$^{7.49}$, or S277$^{7.50}$ (Fig. 4*b*). Note that S$^{3.39}$ is also a highly conserved residue in Class A GPCRs,

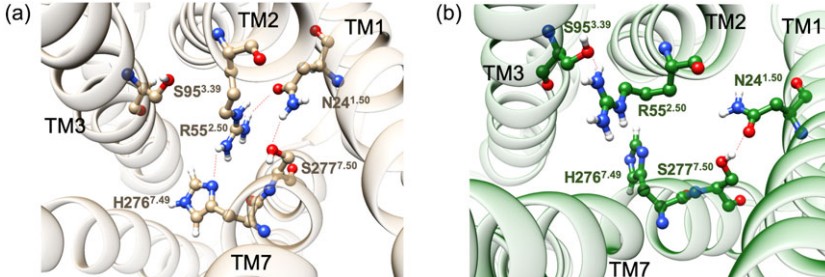

**Fig. 4.** TM1–2-7 interactions (*a*) in the active state and (*b*) in the inactive state, respectively.

where it plays a role in binding sodium together with $D^{2.50}$. In contrast, $S^{3.39}$ is less conserved in TAS2Rs, where it is often mutated to threonine or asparagine (Fig. S3). Since the long side chain of $R55^{2.50}$ can make a direct HB with $S95^{3.39}$ and there is no role for the sodium ion as in Class A GPCRs, there has presumably been less evolutionary pressure at this site in TAS2Rs, since it need only to make an HB with $R55^{2.50}$. Our metaMD studies also indicate that $R55^{2.50}$ prefers to interact with $S95^{3.39}$ rather than $N24^{1.50}$ (Fig. S15). These results indicate that $R^{2.50}$ likely plays an important role in activation, just as does $D^{2.50}$ for typical Class A GPCRs, but differently by stabilizing either the inactive or active state by changing the HB network together with $N^{1.50}$, $S^{3.39}$, $H^{7.49}$ and $S^{7.50}$.

### The DRY motif and $E^{6.30}$

The $D(E)^{3.49}R^{3.50}Y^{3.51}$ motif together with $E(D)^{6.30}$ is another well conserved motif in typical Class A GPCRs. The $R^{3.50} – E(D)^{6.30}$ SB interaction is considered to play a role in maintaining the inactive state, where it is known as the 'ionic lock' (Zhou *et al.*, 2019). For TAS2Rs, however, neither the DRY motif nor $E(D)^{6.30}$ is conserved, and there is no alternative residue at a nearby position to form a SB for the ionic lock. Some Class A GPCRs, such as the opioid receptors, possess an HB that stabilizes the inactive state, instead of a SB, although they still have the DRY motif in TM3.

For TAS2Rs, there are several candidates for this HB: $Y106^{3.50}$ or $K109^{3.53}$ in TM3 and $Q221^{6.30}$ in TM6 (Fig. S16A). Both residues in TM3 are highly conserved, while the 6.30 position is also well conserved with a polar residue (usually serine) for TAS2Rs. However, our metaMD studies on the inactive structure find that none of these pairs are energetically favourable, suggesting that TAS2Rs may not have a TM3-6 interaction stabilizing the inactive state (Figs S16B and S16C). In fact, this possibility of a less stable inactive state or more stable active state of TAS2Rs has been suggested from the significantly different agonist-to-antagonist ratio of TAS2Rs compared to other GPCRs (Di Pizio *et al.*, 2016). Our results support the premise that TAS2Rs have an intrinsically less stable inactive state. This may be because of their roles in protecting animals from harmful compounds, but further experimental studies are needed for validation.

### Conclusion

We used advanced computational techniques to predict the 3D structure of the fully activated TAS2R4 human bitter taste receptor coupled to $G_{gust}$ and to each of several agonists. We found that just as for κOR and μOR, the $G_{gust}$ forms stable SBs with all three ICLs of TAS2R4 while the Gα5 helix makes extensive interactions with residues in the cytoplasmic region of the GPCR. In the binding pocket, $D92^{3.36}$ provides a large binding energy through polar interactions with the agonist, while triggering rotation of $Y239^{6.48}$

by changing the HB network. We find that the unique motifs conserved in TAS2Rs play a role in the activation process but differently compared to typical Class A GPCRs. We also find a less stable inactive state, which may be an intrinsic property of TAS2Rs. This structural information about the fully activated TAS2R4 bitter taste receptor provides new insights that may be useful for understanding of variety of activation mechanisms for other TAS2Rs, providing an insight into the diversity across the GPCR family. These insights should provide guidance useful in developing new therapeutic targets.

### Methods

#### Modelling the human TAS2R4-ligand-G protein complex

The GEnSeMBLE technique (Abrol *et al.*, 2012; Bray *et al.*, 2014) was used to predict the TMD of TAS2R4. The loops were constructed with MODELLER (Fiser *et al.*, 2000), using the mouse μOR structure (PDBID: 6DDF) as a template. DarwinDock (Griffith, 2017) was used to predict the binding site for Rubu and quinine to the TAS2R4 TMD. Initial binding poses of RebM and hydRebM were prepared by matching steviol cores to that of Rubu, followed by 10 cycles of simulated annealing from 50 K to 600 K to optimize the sugar positions. The nucleotide-free gustducin heterotrimeric protein (Guanine nucleotide-binding protein Gt subunit alpha-3: GNAT3, Guanine nucleotide-binding protein Gt subunit beta-1: GNB1 and Guanine nucleotide-binding protein Gt subunit gamma-13: GNG13) was constructed using our optimized $G_iP$ complexed with μOR (Mafi *et al.*, 2020*a*, 2020*b*) as a template to perform homology modelling using Prime (Schrödinger) (Jacobson *et al.*, 2004). The $G_iP$ was chosen because there is a reasonably high similarity (> ~63%) between these two heterotrimeric GP structures. Then, we included the myristoyl-Gly2 in the N-terminus of the GαN helix. Similarly, the inactive $G_{gust}$ bound GDP structure was constructed using the crystal structure of heterotrimeric $G_iP$ bound with GDP (PDBID: 1GOT) as a template for the homology modelling. The initial structure of the TAS2R4-Rubu-$G_{gust}$ complex was modelled by superimposing the predicted human TAS2R4-Rubu with the cryo-EM structure of mouse μOR-$G_i$ (PDBID: 6DDF).

#### System preparation and equilibration

The constructed protein was immersed into 1-palmitoyl-2-oleoyl-sn-glycero-3-phosphocholine (POPC) lipid bilayer, using a pre-equilibrated bilayer structure with 277 molecules (Dong *et al.*, 2016). The protein and lipid membrane were placed in a ~ 100 × 100 × 140 $Å^3$ box, with water molecules and 100 mM concentration of ions (sodium and chloride). The final GPCR-$G_{gust}$ system

contained ∼150,000 atoms. For the inactive structure the system contained 160 POPC with ~70,000 atoms total in a ~ 80 × 80 × 110 Å$^3$ box. We used the Amber14 force field for proteins (Dickson et al., 2014), Ambertools 16 for the lipid (Case et al., 2016) and the generalized Amber force field for the ligands (Wang et al., 2004). Water molecules were described using the TIP3P (Jorgensen et al., 1983) model. Steepest-descent energy minimization was performed first for the constructed system to relax the whole structure. Then, we equilibrated with 5 ns MD simulation, while the positions of all heavy atoms of proteins and ligands were restrained with a force constant of ~2.4 kcal (mol Å$^2$)$^{-1}$ for the first 1 ns, which was then gradually reduced every 1 ns to 0.12 kcal (mol Å$^2$)$^{-1}$. We continued the equilibration with over 200 ns of NPT simulation (and over 100 ns for the inactive structure) without positional restraints to relax the complex structure. During this equilibration step, parts of the helices of TAS2R4, GαN and Gα5 of the Gα subunit, and residues involved in important interactions were often constrained with a force constant of ~1.2 kcal (mol Å$^2$)$^{-1}$ to retain their secondary structures and interactions as the system was relaxed. For free energy analyses, we applied the well-tempered metaMD (Barducci et al., 2008) implemented in PLUMED (Tribello et al., 2014). All simulations were carried out with a 2 fs time step at 310 K and 1 bar using the GROMACS (Abraham et al., 2015) MD software with PLUMED. After the equilibration we carried out 480 ns of MD on Anton2 (Shaw et al., 2014). VMD (Humphrey et al., 1996) and Chimera (Pettersen et al., 2004) programs were partially used for analysis and visualization.

## Human bitterness scores

Pure (~98%) Stevia bitter glycosides were obtained from suppliers (Chromadex Standards, Inc. Los Angeles, CA) or isolated from Stevia leaf according to the procedures of Hellfritsch et al., (2012). Samples were made by dissolving standards in pure sensory-grade water at least 24 h before the test to various levels. Pyschophysical measurements of bitterness intensity were performed as described by Hellfritsch et al. (2012). The 300 ppm level was considered typical for consumption in products such as beverages.

**Acknowledgements.** Funding for this project was provided by Cargill Global Food Research. WAG thanks Dr. Fan Liu for early discussions about TAS2R4 and its activation. The computational resources for this research were provided by the Anton2 computer at the Pittsburgh National Supercomputing Center (MCB180091P) and KISTI National Supercomputing Center (KSC-2018-CHA-0049).

**Open Peer Review.** To view the open peer review materials for this article, please visit http://doi.org/10.1017/qrd.2021.1.

**Supplementary Materials.** To view supplementary material for this article, please visit http://dx.doi.org/10.1017/qrd.2021.1.

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
