## [Reviewer Report]

*Comments to Author*: In this manuscript molecular dynamics simulations using modeled structures of GPCR complexes to describe in atomistic detail interactions that are suggested to be important for the activation of the protein, similarly to what the authors have previously done. In this case the human receptor for bitter taste, TAS2R4, that has been analyzed, Severaltyoes of interactions (mainly hydrogen bonds and salt bridges) that have been identified in other complexes are shown to exist also here. Furthermore the varying degree of bitterness of different compounds is rationalized based on changed interaction patterns.

The simulations are carefully done, although only single simulations have been performed for each system. It is not entirely clear that good equilibration has been obtained fro all systems in the 480 ns MD simulations. For systems of this size (70000-150000) atoms it is only a matter of 1-2 weeks to run a microsecond on a single inexpensive GPU.

Given that there are several variants of the AMBER force fields the authors should provide references for the "Amber 14" and "Ambertools 16" that were used.

---

## [Reviewer Report]

*Comments to Author*: The manuscript "Predicted Structure of Fully Activated Human Bitter Taste Receptor TAS2R4 Complexed with G Protein and Agonists" was submitted by Yang et al. The manuscript was well-written following their previous publication (Mafi et al, PNAS, 2020) with a new insight revealed for the TAS2R4 activation because of its low homology to family A GPCR. In particular, the manuscript proposed a possible differential activation mechanism for Taster receptor, which is interesting and should bring interests for readers to continue studying this type of receptors with this computational model. I would suggest publishing this manuscript with some minor questions.

(1) The authors mentioned that the sequence similarity of TAS2Rs with other class A GPCRS is less than 30% on Page 2, line 8. However, they built up the structures based on A family GPCRs on Page 3, line 12. I am wondering if the author made some justification when they selected these four templates or just as usual for building a model for a high-similarity GPCR. If so, how this justification could be to ensure the built structure maximally close to a natural structure of TAS2Rs, though we don’t have one yet.

(2) On page 10, line 4, the authors mentioned that TAS2Rs neither the DRY motif nor E(D) is conserved, and there is no alternative residue at nearby position to form a SB for the ionic lock. As we know that ionic lock between TM3 and TM6 is in favor of forming inactive states. Is this the reason that TAS2Rs have a less inactive conformation observed.

(3) It is interesting that authors mentioned that for TAS2Rs have several candidate residues forming HB such as Y1063.50 or K1093.53 in TM3 and Q2216.30 in TM6 in which residues in TM3 are more highly conserved in comparison to TM6 residues. Can author make an amino acid alignment for all taste receptors and include some other family A GPCRs to highlight these residues incorporated into Fig. S16 or an independent supplementary figure to give readers a visual view how conserved they are.

(4) It was reported that Gai1-3 have a higher affinity than Ggust, did author even try to use any of Gi to do the simulation and what is different from Ggust engagement in terms of fully activated conformation of the receptor.

---

## [Reviewer Report]

*Comments to Author*: Reviewer #1: The manuscript "Predicted Structure of Fully Activated Human Bitter Taste Receptor TAS2R4 Complexed with G Protein and Agonists" was submitted by Yang et al. The manuscript was well-written following their previous publication (Mafi et al, PNAS, 2020) with a new insight revealed for the TAS2R4 activation because of its low homology to family A GPCR. In particular, the manuscript proposed a possible differential activation mechanism for Taster receptor, which is interesting and should bring interests for readers to continue studying this type of receptors with this computational model. I would suggest publishing this manuscript with some minor questions.

(1) The authors mentioned that the sequence similarity of TAS2Rs with other class A GPCRS is less than 30% on Page 2, line 8. However, they built up the structures based on A family GPCRs on Page 3, line 12. I am wondering if the author made some justification when they selected these four templates or just as usual for building a model for a high-similarity GPCR. If so, how this justification could be to ensure the built structure maximally close to a natural structure of TAS2Rs, though we don’t have one yet.

(2) On page 10, line 4, the authors mentioned that TAS2Rs neither the DRY motif nor E(D) is conserved, and there is no alternative residue at nearby position to form a SB for the ionic lock. As we know that ionic lock between TM3 and TM6 is in favor of forming inactive states. Is this the reason that TAS2Rs have a less inactive conformation observed.

(3) It is interesting that authors mentioned that for TAS2Rs have several candidate residues forming HB such as Y1063.50 or K1093.53 in TM3 and Q2216.30 in TM6 in which residues in TM3 are more highly conserved in comparison to TM6 residues. Can author make an amino acid alignment for all taste receptors and include some other family A GPCRs to highlight these residues incorporated into Fig. S16 or an independent supplementary figure to give readers a visual view how conserved they are.

(4) It was reported that Gai1-3 have a higher affinity than Ggust, did author even try to use any of Gi to do the simulation and what is different from Ggust engagement in terms of fully activated conformation of the receptor.

Reviewer #3: In this manuscript molecular dynamics simulations using modeled structures of GPCR complexes to describe in atomistic detail interactions that are suggested to be important for the activation of the protein, similarly to what the authors have previously done. In this case the human receptor for bitter taste, TAS2R4, that has been analyzed, Severaltyoes of interactions (mainly hydrogen bonds and salt bridges) that have been identified in other complexes are shown to exist also here. Furthermore the varying degree of bitterness of different compounds is rationalized based on changed interaction patterns.

The simulations are carefully done, although only single simulations have been performed for each system. It is not entirely clear that good equilibration has been obtained fro all systems in the 480 ns MD simulations. For systems of this size (70000-150000) atoms it is only a matter of 1-2 weeks to run a microsecond on a single inexpensive GPU.

Given that there are several variants of the AMBER force fields the authors should provide references for the "Amber 14" and "Ambertools 16" that were used.